# Treatment with Paracetamol Can Interfere with the Intradialytic Optical Estimation in Spent Dialysate of Uric Acid but Not of Indoxyl Sulfate

**DOI:** 10.3390/toxins14090610

**Published:** 2022-09-01

**Authors:** Annika Adoberg, Joosep Paats, Jürgen Arund, Annemieke Dhondt, Ivo Fridolin, Griet Glorieux, Jana Holmar, Kai Lauri, Liisi Leis, Merike Luman, Kristjan Pilt, Fredrik Uhlin, Risto Tanner

**Affiliations:** 1Centre of Nephrology, North Estonia Medical Centre, 13419 Tallinn, Estonia; 2Department of Health Technologies, Tallinn University of Technology, 19086 Tallinn, Estonia; 3Nephrology Division, Ghent University Hospital, 9000 Ghent, Belgium; 4Synlab Eesti OÜ, 10138 Tallinn, Estonia; 5Department of Nephrology and Department of Health, Medicine and Caring Sciences, Linköping University, 58185 Linköping, Sweden

**Keywords:** haemodialysis, indoxyl sulfate, optical monitoring, paracetamol, uric acid

## Abstract

Optical online methods are used to monitor the haemodialysis treatment efficiency of end stage kidney disease (ESKD) patients. The aim of this study was to analyse the effect of the administration of UV-absorbing drugs, such as paracetamol (Par), on the accuracy of optical monitoring the removal of uremic toxins uric acid (UA) and indoxyl sulfate (IS) during standard haemodialysis (HD) and haemodiafiltration (HDF) treatments. Nine patients received Par in daily dosages 1–4 g for 30 sessions. For 137 sessions, in 36 patients the total daily dosage of UV-absorbing drugs was less than 500 mg, and for 6 sessions 3 patients received additional UV-absorbing drugs. Par administration slightly affected the accuracy of optically assessed removal of UA expressed as bias between optically and laboratory-assessed reduction ratios (RR) during HD but not HDF employing UV absorbance of spent dialysate (*p* < 0.05) at 295 nm wavelength with the strongest correlation between the concentration of UA and absorbance. Corresponding removal of IS based on fluorescence at Ex280/Em400 nm during HD and HDF was not affected. Administration of UV-absorbing drugs may in some settings influence the accuracy of optical assessments in spent dialysate of the removal of uremic solutes during haemodialysis treatment of ESKD patients.

## 1. Introduction

Optical ultraviolet (UV)-absorbance monitoring of spent dialysate on the outflow from a dialysis machine has become feasibly applied worldwide for assessing the removal of low-molecular-weight uremic solutes from patients’ blood by a haemodialysis procedure [1,2,3,4]. In addition, a strong correlation has been found between optical properties of spent dialysate and concentration of characteristic uremic toxins, such as uric acid (UA) [5,6], indoxyl sulfate (IS) [7], and β-2-microglobulin [8,9]. Nevertheless, there have been indications that the administration of some drug chromophores, e.g., paracetamol (Par), to dialysis patients could disturb the accuracy of the optical methods [10,11]. However, the extent of error caused by drug chromophores in the UV monitoring of dialysis has not been described. Moreover, the dependence of UV measurements on wavelength has not been systematically studied, and the effect of drugs on the fluorescent-based optical monitoring methods have not yet been studied in connection with dialysis treatment of ESKD patients. This study tries to fill these gaps, pointing out the possible interference in the clinical output of optical methods.

## 2. Results

A total of 48 haemodialysis patients (40 male) with a mean age of 63 ± 16 years were studied as detailed in the Material and Methods Section. All patients received medicines according to their prescribed medication plan (cardiovascular medication, potassium and phosphate binders, iron supplementation, analgesics, etc.), including UV (200–400 nm)-absorbing drugs.

To study the effect of the administration of UV-absorbing drugs on the accuracy of the optical monitoring of the removal of uremic toxins, the dialysis sessions were divided into two groups. In the Par− group, any of the prescriptions included UV-absorbing medications in total daily doses <500 mg (137 sessions), and in the Par+ group of dialysis sessions patients additionally received UV-absorbing-drug Par in daily doses >500 mg (30 sessions). In addition, the data of three other chromophoric drugs (ampicillin, flucloxacillin, and valaciclovir) were added for comparison.

### 2.1. Influence of Drugs on the Correlation between Uric Acid and UV Absorbance of the Dialysate

The strongest correlation between the measured concentration of the UA and UV absorbance of spent dialysate samples from the total dialysate collection (‘tank samples’) was found at wavelengths of 294 nm (coefficient of determination *R*^2^ = 0.92) and 292 nm (*R*^2^ = 0.95) for control (Par−, *N* = 137) and Par (Par+, *N* = 30) groups, respectively (Figure 1).

Figure 2 illustrates a representative chromatogram of the spent dialysate of a patient (#1 in Table 3 below) from the Par+ group and the UV-absorbance spectra of Par, Par metabolites, and UA peaks on the insert. The patient received twice 1 g of Par before dialysis and 1 g during the dialysis session and 4 times 1 g on the previous day. Even at the wavelength close to the strongest correlation between the content of UA and absorbance of dialysate, the absorbance of Par main metabolites still remained remarkable (~1/4) compared to that of UA, whereas the contribution of Par and IS to the total absorbance at 295 nm was considerably lower (Figure 2).

There was a strong relationship between the measured concentration of UA in the spent dialysate and UV absorbance of spent dialysate at 295 nm. However, the total absorbance increased by 27.8 ± 10.8% when patients were administered Par (Figure 3). Exceptional 6 episodes where patients received additional UV-absorbing drugs were included for illustrational purposes and not included either into the trendline calculations or statistical analysis shown as follows in Figure 3, Figure 4 and Figure 5.

### 2.2. Influence of Drugs on Correlation between Indoxyl Sulfate and Fluorescence of Dialysate

For the protein-bound uremic toxin IS (excitation at 280 nm), a strong correlation between the measured concentration of IS and the primary-inner-filter effect-corrected fluorescence in spent dialysate samples was found at an emission wavelength of 410 nm (*R*^2^ = 0.89; Par− group; *N* = 137) while the strongest correlation for patients who were administered Par was observed at 385 nm (*R*^2^ = 0.94; Par+ group; *N* = 30; Figure 4).

The correlation between the fluorescence (Ex280/Em400 nm) of spent dialysate samples and measured concentration of IS was comparable in the group of patients who were administered Par (Par+) and the group of patients who were not (Par−; Figure 5).

### 2.3. Influence of Paracetamol on Optical Removal Ratio Monitoring

Mean removal ratio (RR) values of uremic toxins concerned were not statistically different between Par− and Par+ groups, calculated on the basis of optical parameters or laboratory analyses of dialysate samples (Table 1).

The statistical difference between the RR bias values of Par− and Par+ groups was found in the case of UA for haemodialysis (HD) treatment (Figure 6a,b, *p* = 0.045), but not for haemodiafiltration (HDF) treatment (*p* = 0.093). The differences in bias values between Par− and Par+ groups were not statistically significant for IS during both HD and HDF treatments (*p* > 0.6).

## 3. Discussion

This clinical study examined the effect of Par administration on the accuracy of optically monitoring the removal of uremic toxins UA and IS during the haemodialysis of ESKD patients. The main findings of the study were: (1) the administration of chromophoric drug Par in large amounts increased the UV absorbance of spent dialysate, which can lead to the overestimation of concentration and the RR of UA when evaluated by UV-absorbance of spent dialysate, using the UV region that overlaps with the Par-absorption spectrum; (2) fluorescence-based optical methods were not affected by Par intake, when fluorescence intensity was corrected for the primary-inner-filter effect; and (3) conventionally prescribed drugs in connection with dialysis treatment did not interfere with the optical monitoring of the treatment.

The strongest correlation between the concentration of UA in spent dialysate and UV absorption in the Par− (control) group was found at 294 nm, and it coincided with the value previously observed by Jerotskaja et al. [5,6]. This value coincides also with the UA absorbance maximum in a water solution at 294.46 nm [12]. In the Par+ group, the strongest correlation between UA concentration and UV absorbance was only slightly shifted toward a shorter wavelength (292 nm). However, as illustrated in Figure 1, the correlation decreases toward the shorter wavelengths of UV light used for measuring the absorption and, simultaneously, the difference between Par− and Par+ groups increases. These results show that the UV monitoring of dialysis at shorter wavelengths (towards absorbance maximum of Par, Figure 2 insert) is less accurate in terms of UA-concentration monitoring and has some limitations related to UV-absorbing-drug administration. Moreover, Vasquez-Rios et al. recently determined that the monitoring of urea removal during haemodialysis treatment by UV absorbance was not unique for a single substance and may be influenced by the intake of medications, such as Par [13]. The administration of Par may similarly affect urea-removal monitoring near 280 nm, the UV region commonly used in commercial UV monitors [1,2,3,4,13] to evaluate urea removal. Our results that describe the Par effect on UA-removal monitoring raise the importance of this issue and it requires a more detailed analysis.

These limitations could be overcome by using multiparametric optical models [14] that incorporate several UV wavelengths in order to evaluate the removal of UA, and also the urea, or using the UV region, such as 295 nm, to minimise the influence of Par. Nevertheless, even at the wavelength very close to the maximum absorbance of UA, the influence of Par intake on the optical estimation of the concentration of UA in the spent dialysate is notable (Figure 3). This effect also becomes evident as the direct clinical output comparing optically and laboratory-based assessments of RR in the case of HD dialysis modality (Figure 6a versus Figure 6b), though the effect is small. However, in the case of HDF modality, the difference in optically and laboratory-assessed RR values did not reach statistical significance at 295 nm between Par− and Par+ groups. The reason for this might be partly a choice of the wavelength (295 nm) at which the Par effect was minor and the variance in the Par+ group relatively large in case of HDF compared to HD. Meanwhile, the difference in bias between Par− and Par+ groups was statistically significant for both HD and HDF at ≤293 nm (unpublished data). In addition, it is possible that the modality itself affected the differences in the removal dynamics of UA and Par, and thus their contribution to the total UV absorbance of spent dialysate, which was used for the optical estimation of the removal ratio.

While Par administration had a notable effect on the region of Par absorbance in the case of optically assessing the concentration of UA, the dispersion of optical data within Par− and Par+ groups was evidently smaller in comparison to the difference between the groups. This observation may be interpreted as an indication that most of the drugs conventionally used in connection with dialysis treatment did not interfere with the optical monitoring of the treatment. The position of the triangle of UV-absorbing-drug ampicillin (375 mg × 1 before dialysis) in Figure 3 seems to confirm this conclusion.

No substantial interference from the Par intake by patients was observed in the case of the fluorescence-based assessment of the elimination of IS. These results justify correcting the emission intensity considering the primary-inner-filter effect of excitation light by UV-absorbing ingredients in the dialysate [15,16] in the case of dialysis monitoring using the fluorescence of spent dialysate. However, it can be clearly observed in Figure 6 (sections C&D and G&H) that the fluorescence of the spent dialysate still presents a more variable optical parameter compared to UV absorbance, despite the correction of the emission considering the primary-inner-filter effect of the UV-absorbing ingredients in the dialysate. The variability in fluorescence of the spent dialysate can be expected as in addition to IS, tryptophane and its other metabolites have a considerable contribution to the fluorescence of the spent dialysate [7] and different removal kinetics [7,17]. However, the high probability of similarity between Par− and Par+ groups point out the promising outlook of the fluorescence for the optical assessment of the removal of IS during dialysis when evaluating the concentration or total removed solute. Furthermore, this study did not observe any conventionally prescribed drugs that notably affected the accuracy of optical monitoring using fluorescence. The possible role of more metabolites with the indole core, the potential effects of the secondary-inner-filter effect on light emission, and the possible energy transfer (FRET) between different fluorophores [18] are the challenges of further research concerning the usage of fluorescence for dialysis-treatment monitoring.

A limitation of the present study is that there was relatively small number of treatment sessions in the Par+ group in comparison to the Par− group, which may have limited the final conclusion regarding the differences in the RR evaluation of Par− and Par+ groups. In addition, patients in the Par− and Par+ groups did not have the same baseline medication history, which may have had some indirect effect on the levels of chromophoric uremic toxins in ESKD patients. Notwithstanding the relatively limited sample size and the latter, this study offered valuable insights that showed that Par administration could affect the accuracy of the optical monitoring of haemodialysis treatment in the UV region of the Par-absorption spectrum when appropriate optical algorithms were not used. Further research in this field should focus on developing optical models that minimise the effect of UV-absorbing drugs.

## 4. Conclusions

The main finding of this study was that the administration of the chromophoric drug Par in relatively large amounts before and during dialysis sessions affected the correlation between UV absorbance and the content of UA in the spent dialysate. Par can lead to some overestimations of the concentration of UA on the basis of the UV absorbance of spent dialysate and also RR in the case of HD treatments, even when using a wavelength of 295 nm, very close to the maximum absorbance of UA. The correlation between the IS concentration and fluorescence in the spent dialysate is not affected by the administration of Par to dialysis patients, neither is the optical assessment of the RR of IS on the basis of the fluorescence of spent dialysate. Additionally, conventionally used drugs in connection with haemodialysis treatment do not interfere with the optical monitoring of the treatment.

## 5. Materials and Methods

In total, 48 ESKD patients were enrolled into the study: 21 from the Centre of Nephrology at the North Estonia Medical Centre, Tallinn, Estonia; 17 from Linköping University Hospital, in Linköping, Sweden; and 10 from Ghent University Hospital, in Ghent, Belgium. All studies were performed after approval of the study protocol by local ethics committees: Tallinn Medical Research Ethics Committee at the National Institute for Health Development, Estonia, decision no. 2205 (issued 27 December 2017); Linköping Regional Medical Research Ethics Committee, Linköping, Sweden, decision no. 2017/593-31 (issued 17 January 2018); Ghent University Hospital, Commissie voor Medische Ethiek, Ghent, Belgium, decision no. B670201938627 (issued 15 February 2019). Informed consent was obtained from all participating patients. All clinical information and biological samples obtained from the clinical networks were transmitted only after anonymization/blinding.

The inclusion criteria of patients were the following: chronic haemodialysis patients older than 18 years with a life expectancy of more than 6 months, with a vascular access capable to obtain a blood flow of at least 300 mL/min and dialysing 3 times weekly for 4 h. The clinical data of the 48 participants and treatment settings are presented in Table 2.

We studied the effect of the administration of Par to the optical assessment of uremic toxins UA and IS removal during haemodialysis (HD) and haemodiafiltration (HDF) treatments. In addition, the data for three other chromophoric drugs (ampicillin, flucloxacillin, and valaciclovir) were added for comparison. All patients were observed during four midweek sessions, including one haemodialysis (HD) treatment and three different modifications of haemodiafiltration (HDF) treatments with different blood- and dialysate-flow combinations. In total, 19 treatment episodes had to be eliminated from the statistics where sampling or measurement mistakes were observed during data quality-control session, or Par or its metabolites were found in the samples of patients who had not been prescribed Par. All patients received medicines according to their prescribed medication plan (cardiovascular medication, potassium and phosphate binders, iron supplementation, analgesics, etc., including UV (200–400 nm) absorbing drugs. The complete list of medicines that were prescribed to the patients according to their individual medication plan has been shown in Appendix A. Dialysis treatments were divided into two groups based on UV-absorbing-drugs administration. In total, 137 dialysis treatments were included into the Par− group where Par was not prescribed, and any of the prescriptions included UV-absorbing medications in total less than 500 mg per day. Altogether, 30 treatments were included in the Par+ group where Par was additionally prescribed to patients, as described in Table 2 (daily doses > 500 mg). During 6 episodes, patients received additional UV-absorbing drugs in high dosages: ampicillin, flucloxacillin (together with Par) or valaciclovir, which were excluded from the analysis of Par influence. The actual intake of Par by the patients was confirmed by the finding of the Par and its metabolite peaks, paracetamol glucuronide (ParG), and paracetamol sulfate (ParS) in HPLC chromatograms of dialysate samples. Similarly, the absence of Par metabolites was checked in dialysate samples from the Par− group of dialysis treatments. Prescription details of UV-absorbing drugs are presented in Table 3.

During each dialysis session, spent dialysate samples were collected 7 min after the beginning and at the end of sessions. In addition, all the dialysates from each treatment were gathered into a collection tank and mixed before sampling (tank dialysate sample, hereafter).

The UV absorbance and fluorescence of the dialysate samples were measured with spectrophotometer UV-3600 (Shimadzu Corp., Japan) and spectrofluorometer RF-6000 (Shimadzu Corp., Japan), respectively. The results of the fluorescence measurement of dialysate samples were corrected considering the sample’s self-absorption of the exciting light (the primary-inner-filter effect—[15,16]). The concentrations of UA and IS were determined using HPLC equipment of Dionex/Thermo Fisher (USA), as previously described [17]. Briefly, two continuous columns of Poroshell 120 C18 4.6 × 150 mm with a security guard Poroshell 120 C18 4.6 × 3 mm were obtained from Agilent Instruments (Santa Clara, CA, USA). The eluent was mixed with 0.05 M formic acid adjusted to pH 4.25 with ammonium hydroxide (A), and an organic solvent mixture of HPLC-grade methanol and HPLC-S-grade acetonitrile, both from Honeywell (Charlotte, NC, USA) in a ratio of 9:1 with 0.05 M ammonium formate salt (B). The three-step linear-gradient elution program was used with the total flow rate of 0.6 mL/min at the column temperature of 40 °C. Diode array spectrophotometric detector (200–400 nm) and fluorescence detector (Ex280/Em360) were used for signal recording.

Spent dialysate samples from the beginning and end of treatments were used for the calculation of removal rates of UA and IS based on HPLC analyses (referred as ‘lab’ analyses), as well as UV 295 nm (in the case of UA) and fluorescence at Ex280/Em400 nm (IS) measurements (‘opt’ estimations). Linear regression analysis was used to investigate the relationship between optical properties of spent dialysate and concentration of UA or IS in spent dialysate samples. Mean change of optical parameters of dialysate samples of Par+ group in relation to the control group was evaluated. Par-dependent deviation was calculated as the mean of relative deviations (%) between the measured optical parameters of patients of the Par+ group and values that corresponded to the same toxin concentrations calculated from the regression equation of the Par− group of patients:(1)$$\mathrm{Deviation}\;\left(\%\right)=\frac{Y\left(\mathrm{Par}+\right)-\;Y\left(\mathrm{Par}-\right)}{Y\left(\mathrm{Par}-\right)}\cdot 100,$$
where Y(Par+) is the measured UV absorbance or fluorescence intensity of the dialysate for the definite Par+ patient; Y(Par−) is the UV absorbance or fluorescence calculated for the same concentration of UA or IS according to the trendline equation that describes the relationship between toxin (UA or IS) concentrations and corresponding optical parameter (absorbance at 295 nm or fluorescence at Ex280/Em400 nm) for the Par− group of patients.

In addition, linear regression analysis was employed to investigate the relationship between laboratory- and optically estimated concentrations of UA or IS in spent dialysate samples of Par− and Par+ groups. Individual regression equations and statistical metrics were found for Par− and Par+ groups over the optical region of interest.

Dialysate-based removal rates of toxins determined in the lab and optically were calculated as:(2)$$\mathrm{RR}\left(\mathrm{lab}/\mathrm{opt}\right)\;\left(\%\right)=\frac{C\left(\mathrm{start}\right)-\;C\left(\mathrm{end}\right)}{C\left(\mathrm{start}\right)}\cdot 100,$$
where for RR(lab), C(start) and C(end) are the corresponding concentrations of UA or IS in dialysate samples analysed in the laboratory; and for RR(opt), C(start) and C(end) are directly measured UV-absorption rates at 295 nm or fluorescence at Ex280/Em400 nm of spent dialysate correspondingly at the beginning (7th min from the start) and end of the treatment (240th min). A two-tailed two-sample *t*-test, assuming unequal variances, was used for a comparison of RR(lab) and RR(opt) differences between Par− and Par+ groups. A *p*-value of <0.05 was considered significant.

## Figures and Tables

**Figure 1 toxins-14-00610-f001:**
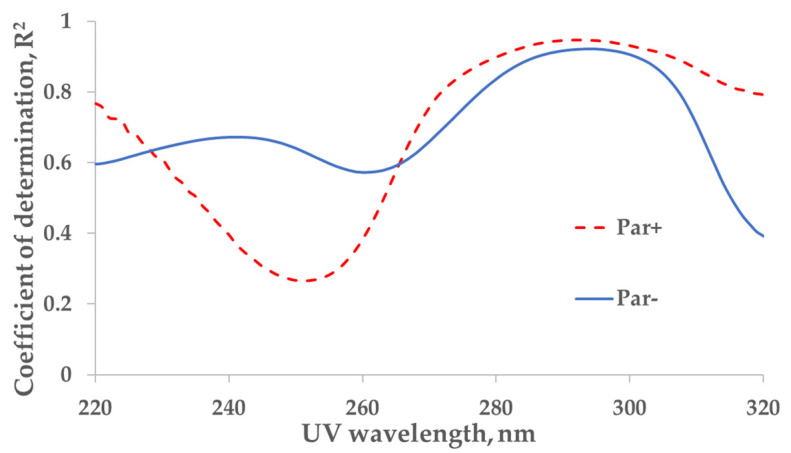
Wavelength dependence of the correlation between UV absorbance of spent dialysate and concentration of uric acid in the spent dialysate. Full line—the control (Par−) group (*N* = 137), determination maximum (*R*^2^) 0.92 at 294 nm. Dashed line—the paracetamol (Par+) group, (*N* = 30), *R*^2^ maximum 0.95 at 292 nm. Local minima can be seen at 260 nm (*R*^2^ = 0.57) and 252 nm (*R*^2^ = 0.27) for control (Par−) and Par+ groups, respectively.

**Figure 2 toxins-14-00610-f002:**
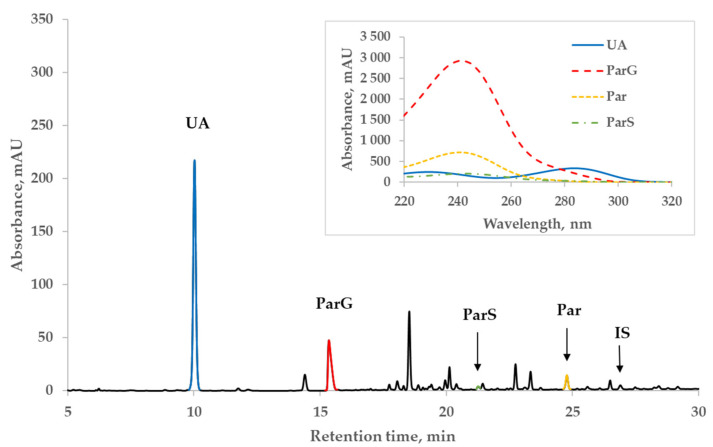
Characteristic HPLC UV 295 nm chromatogram of spent dialysate of patient #1 from the paracetamol (Par+) group. Insert: UV-absorbance spectra of peaks of uric acid (UA), paracetamol glucuronide (ParG), paracetamol (Par), paracetamol sulfate (ParS), and indoxyl sulfate (IS).

**Figure 3 toxins-14-00610-f003:**
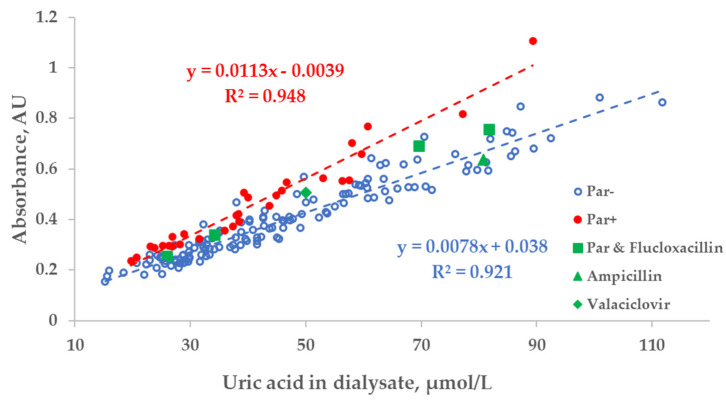
Scatter plot of the measured concentration of uric acid (UA) and UV absorbance at 295 nm of spent dialysate samples. Blue circles—dialysate samples of the control group (Par−, *N* = 137), red dots—group of dialysate samples of patients who received paracetamol (Par+, *N* = 30), green shapes—treatment episodes of patients who received additional UV-absorbing drugs are shown for comparison.

**Figure 4 toxins-14-00610-f004:**
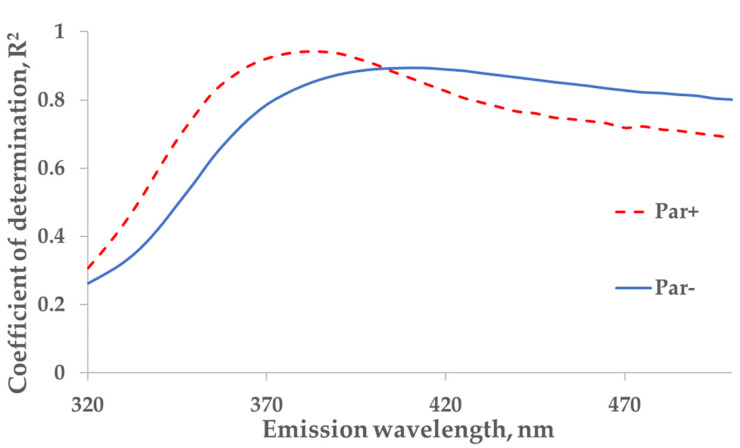
The wavelength dependence of the correlation between the concentration of indoxyl sulfate (IS) and fluorescence in tank samples of spent dialysate, excitation at 280 nm. The strongest correlation between concentration of indoxyl sulfate and emission (Em) are seen at 385 nm for Par+ group (*N* = 30) and Em at 410 nm for Par− group (*N* = 137).

**Figure 5 toxins-14-00610-f005:**
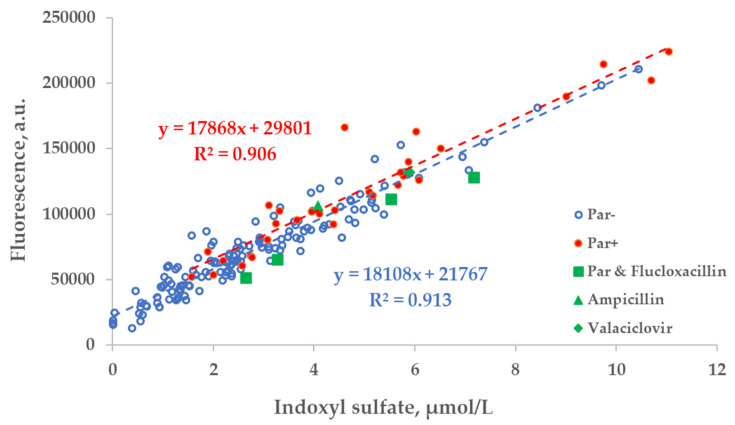
Scatter plot of the concentration of indoxyl sulfate (IS) and fluorescence of spent dialysate samples at Ex280/Em400 nm. Blue circles— control group, (Par−, *N* = 137), red dots— group of patients who received paracetamol (Par+, *N* = 30), green shapes— treatment episodes of patients who received additional UV-absorbing drugs are shown for comparison.

**Figure 6 toxins-14-00610-f006:**
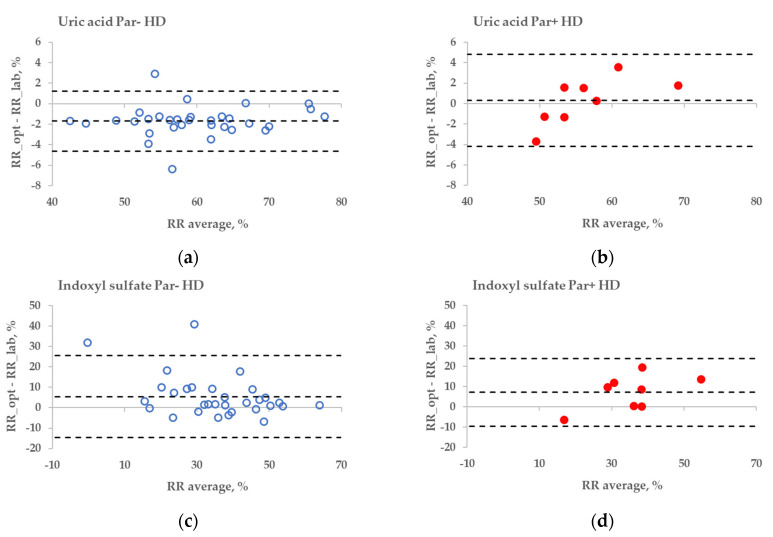
Bland–Altman plot comparing influence of paracetamol on laboratory (lab) and optically (opt) estimated removal ratios (RRs) of uric acid (UA) and indoxyl sulfate (IS) of patients who were administered paracetamol (Par+) or not (Par−) for standard haemodialysis (HD) (**a–d**) and haemodiafiltration modalities (HDF) (**e–h**). UV-absorbance values of spent dialysate at 295 nm were used for estimating RRs of UA and fluorescence Ex280/Em400 for IS.

**Table 1 toxins-14-00610-t001:** Influence of paracetamol on mean removal ratio (RR) values of uric acid (UA) and indoxyl sulfate (IS), calculated on the basis of both laboratory (lab) and optical (opt) parameters of dialysate samples of patients who were administered paracetamol (Par+) or not (Par−) during haemodialysis (HD) and haemodiafiltration (HDF).

Groups Treatments	RR_UA_lab	RR_UA_opt	RR_IS_lab	RR_IS_opt
Par− HD	60.68 ± 8.40	59.00 ± 8.57	32.88 ± 16.30	38.34 ± 12.27
Par+ HD	56.2 ± 5.62	56.52 ± 7.19	31.66 ± 9.09	38.89 ± 13.62
*p*-value	0.091	0.419	0.783	0.919
Par− HDF	75.41 ± 6.88	73.67 ± 6.78	51.92 ± 13.52	54.48 ± 9.77
Par+ HDF	73.37 ± 4.19	72.64 ± 4.23	50.88 ± 10.20	54.33 ± 7.49
*p*-value	0.096	0.396	0.707	0.943

Lab: high-performance liquid chromatography results; opt: results based on UV absorbance at 295 nm (UA) and fluorescence at Ex280/Em400 nm (IS). The statistical comparison: Par+ versus Par− groups.

**Table 2 toxins-14-00610-t002:** Clinical data of the studied end-stage kidney disease patients and treatment settings. Numerical values are presented as mean ± *SD* or median and interquartile range (Q1–Q3.).

Entity of the Data	Specifications		
	Par− Group	Par+ Group	Exceptional Cases *
No. of patients *	38	9	3
Cause of ESKD	Diabetes (4); glomerulonephritis (8); hypertension (10); ADPKD (2); renal carcinoma (3); tubulointerstitial nephritis (5); other (6)	Diabetes (3); glomerulonephritis (1); hypertension (1); ADPKD (1); renal carcinoma (1); other (2)	Diabetes (1); other (2)
Age (years)	62 ± 16	68 ± 15	45 ± 15
Gender	M (32), F (6)	M (7), F (2)	M (2), F (1)
Race	Caucasian 100%	Caucasian 100%	Caucasian 100%
BW ^a^, kg	78.0 (68.0–87.9)	79.0 (74.6–84.1)	79.9 (73.5–89.8)
BMI ^a^, kg/m^2^	25.8 (23.0–29.5)	27.2 (23.3–28.1)	25.8 (24.4–27.3)
Urinary volume ^a^, mL	0 (19 patients) 800 (335–1200) (19 patients)	0 (5 patients) 575 (388–750) (4 patients)	0 (2 patients) 2700 (2700–2700) (1 patient)
Pre-dialysis-serum total protein ^a^, g/L	66.3 (61.0–68.9)	68.0 (66.0–71.0)	68.5 (64.1–71.3)
Pre-dialysis haematocrit ^a^, %	35.8 (33.9–37.7)	38.0 (37.0–39.0)	32.7 (30.4–35.5)
Pre-dialysis-serum calcium ^a^, mmol/L	2.30 (2.18–2.39)	2.31 (2.20–2.36)	2.26 (2.22–2.36)
Pre-dialysis-serum phosphorus ^a^, mmol/L	1.70 (1.22–1.95)	1.50 (1.40–1.66)	2.25 (2.14–2.38)
Dialysis vintage, months	32 (12–89)	51 (25–63)	12 (11–48)
Vascular access	Native fistula (28); graft (8); catheter (2)	Native fistula (8); graft (1)	Native fistula (3)
No. of dialyses	35 (HD) 102 (HDF)	8 (HD) 23 (HDF)	3 (HD) 3 (HDF)
spKt/V	1.04 (0.90–1.17) (HD) 1.60 (1.36–1.83) (HDF)	1.13 (0.94–1.17) (HD) 1.69 (1.52–1.83) (HDF)	1.03 (0.99–1.07) (HD) 1.44 (1.40–1.56) (HDF)
Blood flow (Q_b) effective, mL/min	199 (199–199)) (HD) 300 (297–356) (HDF)	199 (199–200) (HD) 345 (298–360) (HDF)	199 (199–199) (HD) 297 (282–347) (HDF)
Dialysate flow (Q_d), mL/min	299 (297–300) (HD) 789 (500–800) (HDF)	300 (300–300) HD) 800 (497–800) (HDF)	300 (299–300) (HD) 795 (558–798) (HDF)
Ultrafiltration volume, mL	2378 (1051–3336) (HD) 2000 (1336–3000) (HDF)	2050 (1400–2445) (HD) 2500 (2000–2937) (HDF)	4000 (2200–4150) (HD) 398 (397–399) (HDF)
Liquid-substitution volume, L	0 (HD) 22 (15–25) (HDF)	0 (HD) 22 (15–26) (HDF)	0 (HD) 24 (20–25) (HDF)
Dialysis membrane surface area, m^2^	1.5 (1.4–1.5) (HD) 2.2 (2.1–2.2) (HDF)	1.4 (1.4–1.4) (HD) 2.1 (2.0–2.1) (HDF)	1.4 (1.4–1.5) (HD) 1.8 (1.8–1.8) (HDF)

*: Data of 6 exceptional treatments sessions of 3 patients who received ampicillin, flucloxacillin (together with Par), or valaciclovir that were excluded from the analysis of paracetamol influence. ^a^: Assessed during standard treatment prescribed to the patients. Abbreviations: ADPKD—autosomal dominant polycystic kidney disease; M—male; F—female; BMI—body mass index; BW—body weight at the end of the session; spKt/V—single-pool KtV urea, HD—standard haemodialysis, HDF—haemodiafiltration: three different settings with various blood- and dialysate-flow combinations. Dialysator types: HD Xevonta Lo 15 (*N* = 20), FX60 (*N* = 10), Revaclear 300 (*N* = 13); HDF—FX800 (*N* = 36), FX1000 (*N* = 72), Polyflux 210 H (*N* = 16). The effective membrane surface areas of dialysers were the following: FX60 1.4 m^2^, FX800 1.8 m^2^, FX1000 2.2 m^2^, Polyflux 210 H 2.1 m^2^, Revaclear 300 1.4 m^2^, Xevonta Lo 15 1.5 m^2^.

**Table 3 toxins-14-00610-t003:** Prescription of UV-absorbing drugs to end-stage kidney disease patients of the paracetamol (Par+) group.

Patient No.	UV-Absorbing Drug	Daily Dosage
#1	Paracetamol	1 g × 4 or 1 g × 3
#2	Paracetamol	1 g × 4
#3	Paracetamol	1.33 g × 3 or 1 g × 3
#4	Paracetamol	1 g × 4
#5	Paracetamol	1 g × 3
#6	Paracetamol	1 g × 2
#7	Paracetamol	1 g × 4
#8	Paracetamol	1 g × 2
#9	Paracetamol	1 g
#10*	Paracetamol + Flucloxacillin *	1 g × 4 + 0.75 g × 3
#11*	Ampicillin *	0.375 g × 1
#12*	Valaciclovir *	0.5 g × 2

*: Prescribed and indicated in Figure 3 and Figure 5 for illustration, but corresponding dialyses were not included in either Par+ or Par− groups for statistical analyses concerning the influence of paracetamol.

## Data Availability

Data sharing is not applicable due to legal and privacy issues.

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
