# Peer review of "Treatment with Paracetamol Can Interfere with the Intradialytic Optical Estimation in Spent Dialysate of Uric Acid but Not of Indoxyl Sulfate"

_toxins, 2022, doi:10.3390/toxins14090610_

Round 1

Reviewer 1 Report

The author designed this study to assess the effect of the administration of UV-absorbing drugs (paracetamol) on the accuracy of optical monitoring of the removal of uremic toxins (uric acid and indoxyl sulfate) during standard hemodialysis treatment.

There are some concerns with this manuscript. It needs intensive modification to polish it.

Major comments

1.     In the introduction part, the author didn’t need to show the detail of the study design,  which it would be shown in the method part. Moreover, did any research gap exist in this field? As the author mentioned that “paracetamol (Par) to dialysis patients, could disturb the accuracy of the optical methods,” meaning such association already was known. 

2.     In result part, I suggest the author us subtitles to enchaining the readability.

3.     In the result part, the conclusion for Fig3 was not precise because the baseline medication history was not balanced between the groups of Par+ and Par-. 

4.     In table 1, I can’t see any information about RR of IS. Please provide such information and the p-value about the difference between RR_lab and RR_opt.

5.     In figure 6, the authors motioned that the differences of bias just in UA Par- and UA Par+ were statistically significant. However, from the picture, the difference in bias in other groups seems obvious.

6.     In the discussion part, the main finding can’t be supported by the current analysis.

7.     In table 2,  Please provide the detail of the medication history of the participants.

8.     To test the difference between RR(lab) and RR(opt) in this manuscript, the authors should use the method of paired-t-test because it is correlated data. 

Reviewer 2 Report

The paper described the influence of Paracetamol on the optically assessed removal of uric acid in spent dialysate during standard haemodialysis.

The paper is interesting but need some minor revision.

It’s not clear if the same patients are in the two group Par+ and Par-. In the results line 46 it is said that the dialysis sessions were divided in Par+ and par-. In the line 48 we found two groups of patients. I think this point needs to be clarified: groups of patients or groups of sessions with the same patients.

If in the two groups we have the same patients it may be interesting two show the difference in absorbance of spent dialysate in the same patients in a session PAR+ and in a session Par-

The patients No 10, 11, and 12 have to be excluded from the Par+ groups if they were excluded from the analysis of the Par influence.

The KT/V and the Blood flow in the HD sessions are very low (Table 2). This can reflect a vascular access problem. In the methods line 209 authors declared: “vascular access capable to obtain a blood flow of at least 300 mL/min” were included. The authors have to exclude the patient with a low Qb.
